# Umbelliferose Isolated from *Cuminum cyminum* L. Seeds Inhibits Antigen-Induced Degranulation in Rat Basophilic Leukemia RBL-2H3 Cells

**DOI:** 10.3390/molecules27134101

**Published:** 2022-06-25

**Authors:** Momoko Ishida, Rika Ohara, Fuka Miyagawa, Hiroe Kikuzaki, Kosuke Nishi, Hiroyuki Onda, Nanami Yoshino, Takuya Sugahara

**Affiliations:** 1Department of Bioscience, Graduate School of Agriculture, Ehime University, Matsuyama 790-8566, Japan; ishida.momoko.vb@ehime-u.ac.jp (M.I.); fuka19961201@gmail.com (F.M.); nishi.kosuke.mx@ehime-u.ac.jp (K.N.); 2Department of Food and Nutritional Sciences, Graduate School of Humanities and Science, Nara Women’s University, Kitauoya Nishimachi, Nara 630-8506, Japan; sar_ohara@cc.nara-wu.ac.jp; 3Department of Food Science & Nutrition, Nara Women’s University, Kitauoya Nishimachi, Nara 630-8506, Japan; kikuzaki@cc.nara-wu.ac.jp; 4Food and Health Sciences Research Center, Ehime University, Matsuyama 790-8566, Japan; 5Central Research Institute, S&B Foods Inc., #605 Mitsui Link-lab Shinkiba 1, 2-3-8 Shinkiba, Koto-ku, Tokyo 136-0082, Japan; hiroyuki_onda@sbfoods.co.jp (H.O.); nanami_yoshino@sbfoods.co.jp (N.Y.)

**Keywords:** anti-allergic effect, degranulation, *Cuminum cyminum* L., umbelliferose

## Abstract

*Cuminum cyminum* L. (cumin) is an annual plant of the *Umbelliferae* family native to Egypt. We previously showed that the aqueous extract of cumin seeds suppresses degranulation by downregulating the activation of antigen-induced intracellular signaling molecules in rat basophilic leukemia RBL-2H3 cells. However, the active substances in the extract have not yet been identified. Accordingly, herein, we aimed to ascertain the water-soluble substances present in cumin seeds that inhibit degranulation, which led to the identification of umbelliferose, a characteristic trisaccharide present in plants of the *Umbelliferae* family. Our study is the first to reveal the degranulation-suppressing activity of umbelliferose, and quantification studies suggest that cumin seed powder contains 1.6% umbelliferose. Raffinose, an isomer of umbelliferose, was also found to significantly suppress antigen-induced degranulation, but less so than umbelliferose. Both umbelliferose and raffinose contain sucrose subunits in their structures, with galactose moieties bound at different sites. These differences in structure suggest that the binding of galactose to the sucrose subunit at the α1-2 bond contributes to its strong degranulation-inhibiting properties.

## 1. Introduction

A type I allergy is associated with an immediate allergic reaction, with symptoms that appear within a few hours after exposure to the allergen. Typical allergens include food, pollen, mites, and house dust, and immunoglobulin E (IgE) is produced in the body to eliminate the allergen. IgE binds to specific Fc receptors on the surfaces of mast cells and basophils. The re-entering allergen and IgE cross-link on the receptor, resulting in the release of excessive amounts of chemical mediators, such as histamine and leukotrienes, into the extracellular space [1,2] in a process called “degranulation”. The released chemical mediators cause rapid vasodilation and increased levels of vascular permeability, leading to the leakage of plasma components to the exteriors of blood vessels and a decrease in blood pressure [3]. Repeated allergen exposure causes IgE to accumulate, resulting in rapid and severe anaphylaxis.

Many studies into anti-allergic food-derived ingredients and extracts have been reported. So far, catechin derivatives, such as (−)-epigallocatechin-3-*O*-gallate [4,5] and nobiletin, a polymethoxy flavonoid found in *Citrus* [6], have been reported to have anti-allergic activities. These components act through various mechanisms, including directly on intracellular signaling molecules or indirectly via receptors on cell surfaces. While many of the reported anti-allergic substances are fat-soluble components, research into water-soluble substances, such as proteins and sugars, is gaining ground, with flavonoid glucuronides isolated from aqueous spinach extract [7] and β-lactoglobulin [8] reported to exhibit anti-allergic effects. Although the health functions of extracts from various food materials have been comprehensively elucidated, the active substances in many studies have not yet been identified. Considering the cost of commercialization, and that extracts can be ingested at mealtime, they provide effective therapeutic options. However, it is important to clarify the amount of active substance present in an extract, its effectiveness, as well as its mode of action.

*Cuminum cyminum* L. (cumin) is an annual plant of the *Umbelliferae* family native to Egypt. The seeds are strongly aromatic and spicy, and are used in curries and other foods. The seeds contain volatile components, such as cuminaldehyde (4-isopropylbenzaldehyde), γ-terpinene, and β-pinene [9]; these components have been reported to show various health functions [10,11,12]. Moreover, cumin is a food that has received significant attention from health and beauty perspectives. We previously showed that aqueous cumin-seed extracts suppress degranulation by downregulating the activation of antigen-induced intracellular signaling molecules in rat basophilic leukemia RBL-2H3 cells [13]. However, the active substances in the extract have not yet been identified. Accordingly, herein, we aimed to identify the water-soluble substances present in cumin seeds that inhibit degranulation.

## 2. Results and Discussion

Focusing on the water-soluble components present in cumin seeds, fractionation was performed according to the isolation flow chart shown in Figure 1. First, cumin seed powder degreased with dichloromethane was extracted with 70% aqueous acetone to give a 70% aqueous-acetone extract and its residue. The 70% aqueous-acetone extract was subsequently concentrated and then fractionated into a dichloromethane-soluble fraction and a water-soluble fraction by liquid distribution with dichloromethane. The water-soluble fraction exhibited significant inhibition (60.6% inhibition rate at 5.0 mg/mL) when both fractions were examined for their degranulation-suppressing activities (Table 1). Furthermore, the extract was separated into seven fractions (Frs.) by chromatography using an octa dexyl silyl (ODS) column, with Frs. W-2, -3, -4 (water-eluting fractions) and Fr. 6 (50% acetonitrile-eluting fraction) exhibiting suppression. The suppression activities of Frs. W-1 and 7 (100% acetonitrile-eluting component) could not be evaluated due to the extremely low yields obtained.

Fr. 6 was further fractionated by ODS column chromatography (CC), with a 30% acetonitrile-eluting fraction (Fr. 8) and a 50% acetonitrile-eluting fraction (Fr. 9) obtained; both fractions exhibited degranulation-suppressing activities (Table 1). High-performance liquid chromatography (HPLC) identified luteolin from Fr. 8 and apigenin from Fr. 9. Luteolin and apigenin are flavonoids found in many fruits, vegetables, and herbs, for which a variety of biological and pharmacological properties have been reported, including anti-inflammatory [14,15], antioxidant [16], anticancer [17,18,19], and other activities. In addition, the anti-degranulation effects of luteolin and apigenin have already been reported [20,21,22,23]. Therefore, we focused on the isolation and identification of active components contained in the water-eluting fractions.

We next set about determining the active components present in Frs. W-2–4 and identified a trisaccharide in these fractions by electrospray ionization–mass spectrometry (ESI-MS), and various nuclear magnetic resonance (NMR) experiments led to the conclusion that its structure is α-d-galactopyranosyl-(1→2)-α-d-glucopyranosyl-(1↔2)-β-d-fructofuranoside (umbelliferose) (Figure 2a). Umbelliferose has been identified to be present in the roots of *Angelica archangelica* L., a biennial plant of the *Umbelliferae* family, and is characteristically present in plants of this family [24,25,26,27]. Umbelliferose exhibited significant degranulation-suppressive activity, with a 57% inhibitory rate recorded at 15 mM (Figure 2b). Quantification studies suggest that this extract contains 8.3% umbelliferose, from which we estimate that the defatted cumin seeds contain 2.2% umbelliferose, and that cumin seed powder contains 1.6%. Umbelliferose has not previously been reported to be present in cumin seeds, nor has its degranulation-inhibiting properties been reported; hence, this study is the first to demonstrate the anti-allergic effect of umbelliferose in cumin seeds. In addition, raffinose, an isomer of umbelliferose, was found to significantly suppress degranulation with a 15.5% inhibitory rate recorded at 60 mM, but it is less active than umbelliferose (Figure 3a). While raffinose has been reported to be anti-allergic [28,29], it remains unclear whether or not raffinose suppresses degranulation. Furthermore, stachyose and maltotetraose, which are tetrasaccharides, did not exhibit any degranulation-suppressing activity (Figure 3b,c), suggesting that tetrasaccharides show no activity regardless of the constituent sugars.

Monosaccharides (glucose, galactose, mannose, and fructose) did not show any activity (Figure 4a), while disaccharides (sucrose, maltose, and lactose) showed activity (Figure 4b). These results suggest that degranulation-suppressing activity is exhibited when up to three sugar chains are bound. However, treatment of the cells with maltose and lactose reduced cell viability (Figure 4c), suggesting that their degranulation-suppressing activities are due to cytotoxicity. On the other hand, sucrose did not exhibit any significant cytotoxicity, suggesting that sucrose has anti-allergic activity without cytotoxicity. Both umbelliferose and raffinose contain sucrose subunits in their structures, and galactose is bound at different sites. In umbelliferose, galactose and glucose are bound in an α1-2 manner, and in raffinose, they are bound in an α1-6 manner. These differences in structure suggest that the binding of galactose to the sucrose subunit at the α1-2 bond contributes to the strong inhibitory effect of degranulation.

Although umbelliferose was identified as one of the anti-allergic substances in cumin seeds, its mechanism of action has not yet been investigated. Our previous study suggests that cumin seed aqueous extract suppresses antigen-induced degranulation by downregulating phosphatidylinositol 3-kinase/Akt signaling pathways [13]. However, it is unlikely that umbelliferose directly affects the intracellular signal molecules. Umbelliferose may be involved in the antigen–antibody reaction that occurs on the cell surface, the expression or aggregation of IgE receptors, and other receptors on the cell surface.

In this study, the effect of umbelliferose on the release of histamine, which causes increased vascular permeability, vasodilation, and bronchoconstriction in allergic reaction, was not investigated. However, in our previous study, we reported that oral administration of cumin seed aqueous extract suppressed the IgE-mediated enhancement of vascular permeability in passive cutaneous anaphylaxis model mouse [13]. We also confirmed that umbelliferose does not affect the enzymatic activity of β-hexosaminidase itself (data not shown). Our data suggest that umbelliferose suppresses the release of histamine contained in granules as well as this enzyme, but we need to confirm it before conducting in vivo experiments to examine the effect of umbelliferose on actual allergic symptoms. Further basic data in vitro and in vivo are needed to demonstrate the anti-allergic effect of umbelliferose in humans. In further research, we need to examine the effect using clinical samples such as basophils derived from patients with seasonal allergic rhinitis to pollen as performed in [30]. In addition, there are important issues in demonstrating that ingestion of umbelliferose contributes to the prevention and therapy of human type I allergic diseases, including pollinosis and food allergy. Umbelliferose has been generally known as an indigestible oligosaccharide; therefore, we need to demonstrate that orally administered umbelliferose can be absorbed in the gastrointestinal tract and directly affects the allergic reaction. Some studies have been reported that dietary raffinose exerts an anti-allergic effect [28,29]. In addition, Watanabe et al. [28] reported that raffinose was absorbed intact in the gastrointestinal tract, and even though absorbed raffinose is quite a small proportion of orally administered raffinose, it seems to act via a post-absorptive mechanism. Therefore, it is possible that umbelliferose is also absorbed in small amounts from the gastrointestinal tract and acts directly on mast cells and basophils to suppress antigen-induced degranulation. Although further investigations are required to understand the mechanisms behind the action of umbelliferose, this knowledge may provide a useful approach for the prevention and therapy of human allergic disease.

In summary, we found that luteolin, apigenin, and umbelliferose are the anti-allergic substances present in cumin seeds. In particular, umbelliferose has not previously been reported to have an inhibitory effect on degranulation; hence, this study is the first to demonstrate the anti-allergic effect of umbelliferose in cumin seeds. In addition, comparing the structures and activities of various sugars suggests that the structure of umbelliferose is particularly important for suppressing degranulation. On the other hand, umbelliferose is present in small quantities in cumin seeds, which suggests that interactions between substances are possibly responsible for the observed strong activities of cumin seed aqueous extracts. The water-soluble cumin-seed components need to be further analyzed to obtain better insight into their anti-allergic properties.

## 3. Materials and Methods

### 3.1. Reagents

Dulbecco’s modified Eagle medium (DMEM), penicillin, streptomycin, fetal bovine serum (FBS), bovine serum albumin (BSA), mouse anti-dinitrophenyl (DNP), monoclonal IgE, DNP–human serum albumin (HSA) conjugate, Triton X-100, and deuterated water (D_2_O) were acquired from Sigma-Aldrich (St. Louis, MO, USA). All other chemicals were purchased from Fujifilm Wako Pure Chemical (Osaka, Japan) or Nacalai Tesque (Kyoto, Japan), unless otherwise noted.

### 3.2. General Experimental Procedures

HPLC was performed with a JASCO PU-2080 plus Intelligent Pump (Tokyo, Japan) equipped with a JASCO MD-4015 photo diode array detector or a JASCO PU-1580 intelligent pump equipped with a JASCO RI-1530 intelligent RI detector. NMR spectra were recorded on a Bruker AV300N instrument (300 or 700 MHz, Bruker, MA, USA). LC-MS was performed using an ACQUITY UPLC system and an ACQUITY TQD mass spectrometer (Waters Corporation, Milford, MA, USA) with an ESI source operated in positive mode. ESI conditions were a capillary voltage of 2000 V, a cone voltage of 20 V, a source temperature of 140 °C, and a cone gas (N_2_) flow of 50 L/h. The solvent was 50/50 *(v*/*v*) methanol/water with 0.05% formic acid, at a flow rate of 0.1 mL/min.

### 3.3. Extraction

Powder of cumin seeds imported from Turkey was provided by S&B Foods Inc. (Tokyo, Japan). The powder (200 g) was extracted with dichloromethane (630 mL × 4, 20 h each) at room temperature. Each extract was combined, filtered using Whatman No. 2 paper, and concentrated under reduced pressure to yield the dichloromethane extract (52.7 g) and defatted cumin (residue) (142.8 g). The defatted cumin (136.8 g) was extracted with 70% aqueous acetone (630 mL × 4, 20 h each) at room temperature. Each extract was combined, filtered, and concentrated in vacuo, and the resulting aqueous residue was partitioned three times with the same volume of dichloromethane to yield the dichloromethane-soluble fraction (1.5 g). The residual aqueous solution was lyophilized to give the water-soluble fraction (20.9 g).

### 3.4. Fractionating the Water-Soluble Fraction

The water-soluble fraction (5.0 g) was subjected to Chromatorex ODS (100–200 mesh, Fuji Silysia Chemical Ltd., Aichi, Japan) gel CC (2.0 cm diameter × 20 cm height) to give seven fractions. Frs. W-1-4 were eluted with distilled water (D.W.) (Fr. W-1: 1.0 mg [0.02%], Fr. W-2: 498.4 mg [9.97%], Fr. W-3: 1265.0 mg [25.30%], Fr. W-4: 1341.7 mg [26.83%]). Then, Fr. 5 (33.1 mg [0.66%]) was eluted with 10% acetonitrile, Fr. 6 (1363.5 mg [27.27%]) with 50% acetonitrile, and Fr. 7 (15.0 mg [0.3%]) with 100% acetonitrile.

### 3.5. Isolating and Identifying the Components in the Water-Soluble Fraction

Fr. W-3 of the water-soluble fraction (1.2 g) was subjected to Sephadex LH-20 (GE Healthcare, Buckinghamshire, UK) gel CC (1.2 cm diameter × 23 cm height), with eight fractions obtained by elution with D.W. The third fraction (88 mg) was further divided into three fractions by preparative HPLC (Column: Mightysil NH_2_ (5 µm, 250 × 4.6 mm, Kanto Chemical Co., Inc.); solvent: 70% acetonitrile; flow rate: 1.0 mL/min; detection: refractive index (RI)-1530). The sample was dissolved with D.W. at a concentration of 30 mg/mL. The sample solution (70 µL) was injected. Preparative HPLC was performed 39 times, and the combined second fraction (12 mg) was identified as umbelliferose by ESI-MS and various NMR experiments (Appendix A). ESI-MS (positive) spectrum showed [M+K]^+^ at *m/z* 543.4 (Figure 5a). The NMR data completely correspond with those reported in the literature for the authentic sample [31].

Fr. 6 of the water-soluble fraction (901.2 mg) was subjected to ODS gel CC (1.5 cm diameter × 20 cm height). Fr. 8 (54.0 mg) was eluted with 30% acetonitrile and Fr.9 (10.6 mg) was eluted with 50% acetonitrile, and were found to contain luteolin and apigenin, respectively, based on HPLC and ^1^H-NMR spectroscopy [32,33].

### 3.6. Quantifying Umbelliferose

Umbelliferose was quantified by analytical HPLC using a Mightysil NH_2_ column (5 µm, 250 × 4.6 mm, Kanto Chemical Co., Inc., Tokyo, Japan), with 70% acetonitrile as the solvent (flow rate: 1.0 mL/min) and RI detection. To determine the most efficient solvent for extraction of umbelliferose, each portion of defatted cumin powder (20 mg) was extracted with 1 mL of D.W., 25% acetonitrile, 50% acetonitrile, 75% acetonitrile, and acetonitrile. After sonication for 1 min, each extract solution (10 µL) filtered with 0.45 µm membrane filter was injected in HPLC. As a result, 75% acetonitrile was found to be the most efficient solvent. For quantifying umbelliferose, 20 mg of the defatted cumin powder was extracted with 1 mL of 75% acetonitrile using ultrasonic for 1 min. The ultrasonic extraction was performed 5 times, the combined extract was filtered and evaporated in vacuo for removing acetonitrile, and the residue was freeze-dried to afford a 75% acetonitrile extract (5.3 mg). The freeze-dried extract (1.6 mg) was dissolved in 200 µL of 70% acetonitrile, and 50 µL of the solution was injected in HPLC. Umbelliferose was detected at Rt 23.5 min (Figure 5b). The calibration curve for umbelliferose in the 25–100 µg/mL range was determined: y = 11.119x – 65.441 (R^2^ = 0.9963).

### 3.7. Preparation of Sugars

Maltotetraose and lactose were purchased from Hayashibara Co., Ltd. (Okayama, Japan) and Nacalai Tesque, respectively. Other sugars were purchased from Fujifilm Wako Pure Chemical. All sugars were dissolved in D.W., adjusted to a pH of 7.4, and sterilized using a 0.22 μm membrane filter.

### 3.8. Cells and Cell Cultures

RBL-2H3 cells were obtained from the American Type Culture Collection (Rockville, MD, USA) and cultured in DMEM supplemented with 100 U/mL of penicillin, 100 µg/mL of streptomycin, and 10% FBS at 37 °C in an incubator under humidified 5% CO_2_ [13].

### 3.9. β-Hexosaminidase Release Assay

RBL-2H3 cell degranulation was evaluated by determining the activity of β-hexosaminidase stored in granules and released extracellularly by stimulation with an antigen, as previously described [13]. Briefly, RBL-2H3 cells were seeded in a 96-well culture plate (Corning, NY, USA) at 5.0 × 10^5^ cells/well suspended with 10% FBS-DMEM containing 50 ng/mL of anti-DNP IgE and cultured for 18 h at 37 °C. Following sensitization, the anti-DNP IgE-sensitized cells were washed twice with modified Tyrode’s (MT) buffer (20 mM HEPES, 135 mM NaCl, 5 mM KCl, 1.8 mM CaCl_2_, 1 mM MgCl_2_, 5.6 mM glucose, and 0.05% BSA, pH 7.4) to remove excess anti-DNP IgE. A 120 μL aliquot of MT buffer containing various concentrations of sample (or D.W. as the control) was added to the cells. After incubating for 10 min at 37 °C, 10 μL of 6.25 ng DNP-HSA was added to each well and incubated for 30 min at 37 °C. The supernatant was then transferred from each well to a fresh plate, and 0.1% Triton X-100–MT buffer was added to the cells for 1 h. Both the supernatant and cell lysate were transferred to a fresh 96-well microplate and preincubated for 5 min at 37 °C. The substrate solution (3.3 mM *p*-nitrophenyl-2-acetamido-2-deoxy-β-d-glucopyranoside dissolved in 0.1 M citrate buffer; pH 4.5) was then added to each well and incubated for 25 min at 37 °C. The enzyme reaction was terminated by the addition of 2 M glycine buffer (pH 10.4), after which the absorbance at 415 nm was measured using an iMark microplate reader (Bio-Rad Laboratories, Hercules, CA, USA). In addition, 2 M glycine buffer was added to a blank sample prior to the addition of the substrate solution. The β-hexosaminidase release rate was calculated: [(A_supernatant_ − A_blank of supernatant_)/{(A_supernatant_ − A_blank of supernatant_) + (A_cell lysate_ − A_blank of cell lysate_)}] × 100, where “A” is the absorbance measured for each well.

### 3.10. Cell Viability

Cytotoxicity was examined using the Nacalai Tesque WST-8 Cell Count Reagent in accordance with the manufacturer’s instructions. RBL-2H3 cells were seeded in a 96-well culture plate (Corning, NY, USA) at 5.0 × 10^5^ cells/well suspended with 10% FBS-DMEM containing 50 ng/mL of anti-DNP IgE and cultured for 18 h at 37 °C. Anti-DNP IgE-sensitized cells were treated with various concentrations of sample and stimulated with DNP-HSA as described above. After washing with DMEM, fresh medium containing 10% Cell Count Reagent SF was added to each well, and the plate was incubated at 37 °C for 20–40 min, after which absorbance at 450 nm was measured using the abovementioned iMark microplate reader.

### 3.11. Statistical Analysis

The obtained data were expressed as means ± standard errors of the mean (SEMs). GraphPad Prism version 8.4.3 (GraphPad Software, San Diego, CA, USA) was used to analyze the results using one-way analysis of variance followed by Dunnett’s post-hoc test. Values with * *p* < 0.05, ** *p* < 0.01, and *** *p* < 0.001 were considered statistically significant.

## Figures and Tables

**Figure 1 molecules-27-04101-f001:**
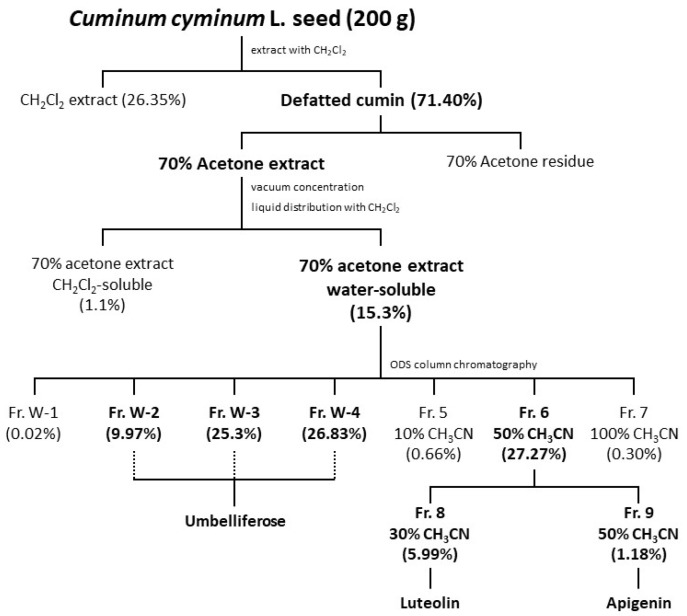
Flow chart for the isolation of active compounds from cumin seeds.

**Figure 2 molecules-27-04101-f002:**
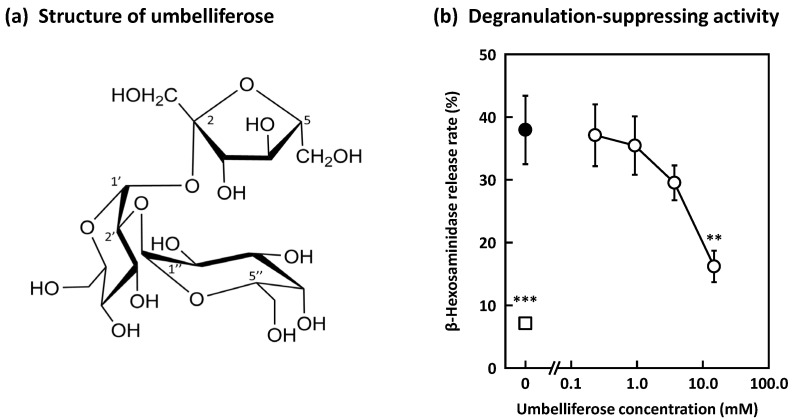
Effect of umbelliferose on RBL-2H3 cell degranulation. (**a**) Structure of umbelliferose. (**b**) Anti-dinitrophenyl (DNP) IgE-sensitized RBL-2H3 cells were treated with various concentrations of umbelliferose (open circles) and induced degranulation by DNP–human serum albumin (HSA) stimulation. The control cells (closed circle) and blank cells (open square) were treated with distilled water, and the control cells were stimulated with DNP-HSA. Data are expressed as means ± SEMs of three independent experiments. ** *p* < 0.01 and *** *p* < 0.001 against the control by Dunnett’s post hoc test.

**Figure 3 molecules-27-04101-f003:**
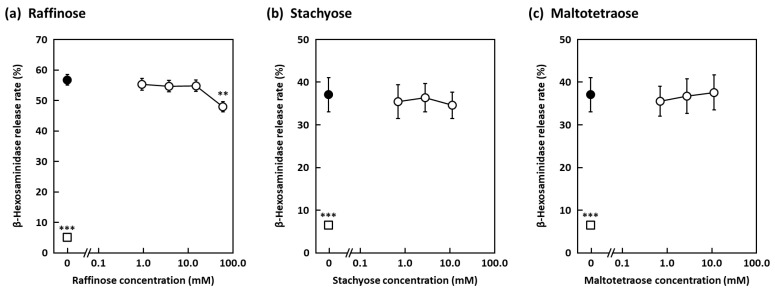
Effects of raffinose, stachyose, and maltotetraose on RBL-2H3 cell degranulation. Anti-dinitrophenyl (DNP) IgE-sensitized RBL-2H3 cells were treated with various concentrations of (**a**) raffinose, (**b**) stachyose, or (**c**) maltotetraose, with degranulation induced by DNP–human serum albumin (HSA) stimulation. Open circles indicate sample-treated cells. The control cells (closed circles) and blank cells (open squares) were treated with distilled water, and the control cells were stimulated with DNP-HSA. Data are expressed as means ± SEMs of three independent experiments. ** *p* < 0.01 and *** *p* < 0.001 against the control by Dunnett’s post hoc test.

**Figure 4 molecules-27-04101-f004:**
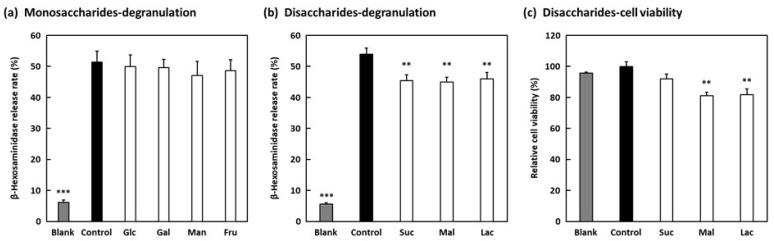
Effects of monosaccharides and disaccharides on RBL-2H3 cell degranulation. (**a**) Anti-dinitrophenyl (DNP) IgE-sensitized RBL-2H3 cells were treated with 50 mM of monosaccharides with degranulation induced by DNP–human serum albumin (HSA) stimulation. Glc, Gal, Man, and Fru indicate glucose, galactose, mannose, and fructose, respectively. The control cells and blank cells were treated with distilled water, and the control cells were stimulated with DNP-HSA. Data are expressed as means ± SEMs of three independent experiments. (**b**) Suc, Mal, and Lac indicate sucrose, maltose, and lactose, respectively. Data are expressed as means ± SEMs of four independent experiments. (**c**) Anti-DNP IgE-sensitized RBL-2H3 cells were treated with 50 mM of disaccharides with degranulation induced by DNP-HSA stimulation, after which cell viability was measured using WST-8 solution. Data are expressed as means ± SEMs of two independent experiments. ** *p* < 0.01 and *** *p* < 0.001 against the control by Dunnett’s post hoc test.

**Figure 5 molecules-27-04101-f005:**
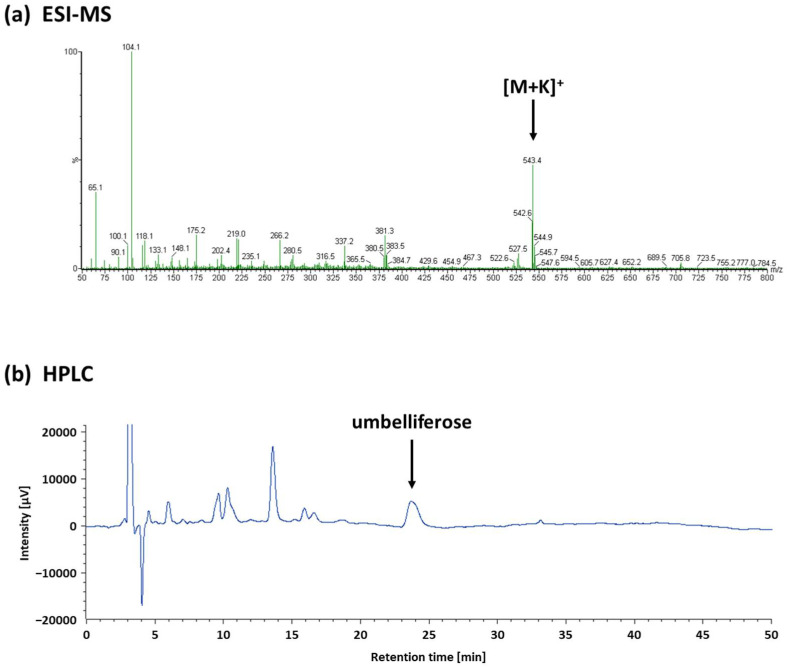
Chemical profiling of defatted cumin seed extract. (**a**) Electrospray ionization–mass spectrometry (positive) *m/z* 543.4 [M+K]^+^. (**b**) Umbelliferose was quantified by high-performance liquid chromatography (Rt 23.5 min) using a Mightysil NH_2_ column (5 µm, 250 × 4.6 mm), with 70% acetonitrile as the solvent (flow rate: 1.0 mL/min) and refractive index detection.

**Table 1 molecules-27-04101-t001:** Extraction data.

Sample	Solvent	Activity	Identified Compound
70% Acetone extract water-soluble fraction	D.W.	+	
Fr. W-1	D.W.	Unrated	
Fr. W-2	D.W.	+	Umbelliferose
Fr. W-3	D.W.	+	
Fr. W-4	D.W.	+	
Fr. 5 10% CH_3_CN eluate	DMSO	−	
Fr. 6 50% CH_3_CN	DMSO	+	
Fr. 7 100% CH_3_CN	DMSO	Unrated	
Fr. 8 30% CH_3_CN eluate	DMSO	+	Luteolin
Fr. 9 50% CN_3_CN eluate	DMSO	+	Apigenin

Samples with inhibition rate of 10% or more are indicated with “+”. D.W. and DMSO indicate distilled water and dimethyl sulfoxide, respectively.

## Data Availability

The data that support the findings in this study are available from the corresponding author upon reasonable request.

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
