# Peer review of "Umbelliferose Isolated from Cuminum cyminum L. Seeds Inhibits Antigen-Induced Degranulation in Rat Basophilic Leukemia RBL-2H3 Cells"

_molecules, 2022, doi:10.3390/molecules27134101_

Round 1
Reviewer 1 Report
This currently submitted MS entitled "Umbelliferous isolated from Cuminum cyminum L. seeds inhibits antigen-induced degranulation in basophilic leukemia RBL-2H3 cells" by Ishida et al. is very interesting article on the isolation and purification of key active principals from cumin seeds by HPLC/NMR/ESI-MS and the screening of the obtained fraction for their abilities to inhibit granulation in rat RBL-2H3 cells. Authors identified umbelliferous as a potent anti-allergic bioactive compound by use of β-Hexosaminidase as degranulation marker. The work experimental design is meticulously planned and authors have provided experimental details in them ethology, design is sound, results and discussion are properly described. However, I have some issues and comments that need to be addressed before considering the MS any further.
MAJOR
-The diagnostic and therapeutic potential of the rat RBL-2H3 cell line is well established. However, it would have been really very informative to show the degranulation-inhibiting properties of the umbelliferous. However, have the authors also determined histamine, which is released concomitantly with β-Hexosaminidase when the cells are immunologically activated?
- It would be important to conduct this study on clinical samples of basophils derived from patients with seasonal allergic rhinitis to pollen. Please cite the below study to list an example of applied aspect of their discovery.
https://pubmed.ncbi.nlm.nih.gov/16465400/
-Authors need to draw a hypothesis on the proposed mode-of-action of umbelliferous , preferably by using a schematic presentation if possible.
- Authors need to elaborate in the discussion further to clarify the novelty of the observed degranulation-inhibiting activity of umbelliferous and its potential applications and perhaps giving some examples of similar natural products that were formulated to tackle food allergens issues.
MINOR:
-I suggest to add the word "rat" before the word "basophilic" in the tittle.
- The origin of the cumin seeds should be mentioned; was it purchased from the local Japanese Markert or imported from Egypt. This will enable future comparative studies of cumin seeds from different geographical locations.
-Acronyms such as ESI-MS and NMR should be written in full at the first apparition, then abbreviated.
-Supplementary Figures 2 and 3 should be included in the main text an merged as one figure titled: Chemical profiling of defatted cumin seed extract, A: HPLC; B: ESI-MS.
- Section 3.10. Cell viability, should be expanded to include the model of the plate reader as well as detailed WST-8 procedure and cited reference.
-In fact, all the Methods, except for 3.9, require proper citation even if the protocol is their own. Proper citations are necessary.
-L55: perhaps it's better to replace the word "sequentially" be "comprehensively".
-All figure captions, need the full name of DNP-HSA.
Author Response
Dear Reviewer
Thank you very much for reviewing our manuscript and offering valuable advice.
We have addressed your comments with responses and revised the manuscript accordingly.
In addition, we regret some mistakes in Supplementary Figure (Figure S1).
We will send you a revised figure.
Please see the attachment.

Reviewer 2 Report
Umbelliferose isolated from Cuminum cyminum L. seeds inhibits 2 antigen-induced degranulation in basophilic leukemia RBL-2H3 3 cells
This is a very interesting paper, I found it very well researched and clearly written, a logic follow-up from the same group’s paper Cytotechnology 2019, 71, 599–326 609.
After the detection and correction of minor typos, it is ready for publication in Molecules
Author Response
Dear Reviewer
Thank you very much for reviewing our manuscript and offering valuable advice.
We regret some mistakes in Supplementary Figure (Figure S1).
We will send you a revised figure.
Please see the attachment.

Reviewer 3 Report
This paper contains valuable information and deserves to be published after some clarifications. It is demonstrated that umbelliferose inhibits antigen-induced degranulation in basophilic leukemia RBL-2H3 3 cells. Riffinose was also shown to posses the activity in the same in vitro tests. What is the significance of obtained results given tha fact that these trisaccharides are non-digestible in humans? Please provide sufficients details in materials and methods section to allow others to repeat your experiments.
Author Response

(The authors gave the same response as above.)
